# Disparities of Health Program Information Systems in Indonesia: A Cross-Sectional Indonesian Health Facility Research 2019

**DOI:** 10.3390/ijerph20054384

**Published:** 2023-03-01

**Authors:** Sri Idaiani, Harimat Hendarwan, Maria Holly Herawati

**Affiliations:** 1Research Centre for Preclinical and Clinical Medicine, National Research and Innovation Agency, Cibinong Science Center, Jalan Raya Jakarta-Bogor Km. 46, Kec. Cibinong, Kabupaten Bogor 16915, West Java, Indonesia; 2Research Centre for Public Health and Nutrition, National Research and Innovation Agency, Cibinong Science Center, Jalan Raya Jakarta-Bogor Km. 46, Kec. Cibinong, Kabupaten Bogor 16915, West Java, Indonesia

**Keywords:** health information system, community health center, disparity, program, application

## Abstract

Although a recording and reporting format for health centers already exists for Indonesia’s standard information system, numerous health applications still need to meet the needs of each program. Therefore, this study aimed to demonstrate the potential disparities in information systems in the application and data collection of health programs among Indonesian community health centers (CHCs) based on provinces and regions. This cross-sectional research used data from 9831 CHCs from the Health Facilities Research 2019 (RIFASKES). Significance was assessed using a chi-square test and analysis of variance (ANOVA). The number of applications was depicted on a map using the spmap command with STATA version 14. It showed that region 2, which represented Java and Bali, was the best, followed by regions 1, which comprised Sumatra Island and its surroundings, and 3, Nusa Tenggara. The highest mean, equaling that of Java, was discovered in three provinces of region 1, namely, Jambi, Lampung, and Bangka Belitung. Furthermore, Papua and West Papua had less than 60% for all types of data-storage programs. Hence, there is a disparity in the health information system in Indonesia by province and region. The results of this analysis recommend future improvement of the CHCs’ information systems.

## 1. Introduction

The established health system needs a robust health information system. One measure of its success is equitable distribution. In the current era of digitalization, there are demands for data to be brought together for both individuals and health facilities [1,2,3]. The availability of data and good monitoring enhance their utilization. Hence, they can replace population-based surveys [4].

Data in health facilities of developed countries are inputted electronically, either online or offline. However, in low-and-middle-income countries (LMICs), entry of health data, especially for primary services, is not adequate [3,5]. Furthermore, other problems include the use of various applications, varied program data requirements, and poor data quality. This is due to lack of monitoring and evaluation of the information system used [6].

Previous research on Indonesian health information systems has covered hospital information systems, electronic medical records, information systems, mHealth, e-health, telemedicine, and the primary healthcare information system (SIMPUS) [7]. Conversely, in developed countries, research on health information systems has included software, artificial intelligence, clinical decision support systems (clinical DSSs), epidemiological monitoring, data mining, patient safety and outcomes, meaningful use, quality improvement, and social media [8].

As one of the LMICs, Indonesia has a diverse primary care information system. Despite a standard information system already being in place—namely, the existence of records and reports known as the Puskesmas (community health centers) Management Information System (SIMPUS), and all its versions—many health programs, such as maternal, dengue, and other ones, have unique applications [6,9]. Finally, health centers do not work on only one type of system or application that contains various management, facility, and individual data. They have an additional burden of running more than 10 required programs.

The expected outcome of the current vision for data includes efficient and high-quality information. Furthermore, the ability of data entry in CHCs varies depending on the number of employees, the ability of the human resources, the electronic devices, and organizational support, such as the available Internet network [10]. These conditions lead to disparities between regions and provinces, which ultimately affect the health information system. This disparity is likely to occur in Indonesia because this country consists of many islands with unequal resources. By examining the existing information systems in primary health services in Indonesia, it is possible to capture the disparities that occur. Demonstrating the disparities is needed to improve health systems [11]. Therefore, this study was conducted to demonstrate the potential disparities of the health information systems in Indonesia, namely, the application and data collection of programs at the CHCs by province and region.

## 2. Materials and Methods

### 2.1. Design

This was an analysis of the Indonesian Health Facility Research (RIFASKES) data for 2019 [12]. The RIFASKES design is a survey. The original survey collected data on health facilities, including the characteristics of facilities, resources, management, organization, planning, supporting facilities, and information systems. Systems consisted of CHCs, hospitals, and other health facilities. Furthermore, the data of this study were related to research on CHCs. The cross-sectional design of the RIFASKES determined the total sample, namely, all the CHCs in Indonesia. The inclusion criteria were registered with the Ministry of Health data in July 2018 and verified by the local District Health Office (DHO). Furthermore, the data of this study focus on CHCs.

### 2.2. Time, Place, and Data Collection

Data were collected from April to May 2019 by teams of two members from five CHCs per team. Each team visited the CHC health center for four days to obtain all the survey information. The enumerator criteria were a minimum college education (Diploma III at the Department of Health), <45 years old, a non-civil servant, were currently not studying, resided in the province of the research location, applied to become an enumerator, and participated in the selection and technical research training.

Data were acquired through interviews with the program manager for the health information system at the CHC, usually the director. Enumerators performed the interviews using a standardized questionnaire. The assessed disease data application is an application installed on the CHC computer. If they had it, we asked if it was used. Additionally, the enumerators observed and searched the documents and applications. This survey did not assess the process and quality of the data.

### 2.3. Questionnaire

The focus of this analysis was on the CHC record system. Three main parts were analyzed: the Ministry of Health application, an application to support health insurance, and an application for health and disease programs. Questions included the availability of 13 types of records, namely: SKDR (Sistim Kewaspadaan Dini dan Respon), or Early Warning of Epidemic Diseases; ASPAK (Aplikasi Sarana Prasarana dan Alat Kesehatan), or Health Facilities and Supplies Application; PISPK (Program Indonesia Sehat Pendekatan Keluarga), or the Healthy Family Program; P Care (Primary care), a health insurance application; HFIS (Health Financing Information System), for financing of health insurance; SITT (Sistim Informasi Tuberculosis Terpadu), for tuberculosis information; SIHA (Sistim Informasi HIV AIDS), for HIV AIDS information; SIHEPI (Sistim Informasi Hepatisis), or Hepatitis Information; SIPTM (Sistim Informasi Penyakit tidak Menular), or Non Communicable Disease Information; SIPD3I (Sistim Informasi Penyakit yang Dapat Dicegah dan Diobati dengan Imunisasi), or Diseases Prevented and Cured by Immunization; ESISMAL (Electronic System oh Malaria), or Malaria Information; SISTBM (Sistim Informasi Sanitasi Total Berbasis Masyarakat), or Health Sanitation Population-Based; EPPGBM (Electronic Pencataan dan Pelaporan Gizi Berbasis Masyarakat), or Nutrition Information.

The information system owned by the CHCs included medical records and SIMPUS. This analysis excluded health program applications that were relatively new and had not been widely used, such as the dengue and mental health applications.

According to a decree issued by the regional government, criteria for CHCs were established based on urban, rural, remote, and very remote areas. Furthermore, the outpatient and inpatient criteria, accreditation status of the CHC, and financial management were determined based on the available document. Division of the territory into seven regions was performed according to the classification of Indonesia’s development areas, as written in the Medium-Term Development Plan [13]. Regions 1, 2, 3, 4, 5, 6, and 7 were provinces located on the island of Sumatra and its surroundings, Java and Bali, Nusa Tenggara, Kalimantan, Sulawesi, Maluku, and Papua, respectively.

### 2.4. Data Management and Analysis

The data processing of RIFASKES 2019 consisted of two stages. The first was conducted in regencies/cities and consisted of data collection, receiving–batching (acceptance–bookkeeping), editing (data quality control), data entry, and sending electronic data. The second was performed in the National Institute of Health Research and Development (NIHRD). It consisted of receiving and merging data for all regencies/cities, cleaning data, merging provincial and national data, cleaning national data, imputation, weighting, and storing electronic data.

The team submitted a data request to the Repository of NIHRD, namely, the data-related health information system at a CHC. Data were tabulated, and the significance was assessed using a chi-square test. Furthermore, the number of applications was assessed for the mean, and analysis of variance (ANOVA) determined the significance. Next, data were mapped using the spmap command with STATA version 14. The categorization was based on the default STATA, which automatically divided the data into four quartiles.

## 3. Results

The CHCs originated from 34 provinces, which included all Indonesian districts and cities. The total CHSs was 9909. We coordinated with local DHOs before the study was implemented. As a result, 14 health centers were eliminated, since they were declared unavailable. There were 9885 CHCs visited; furthermore, 54 changed their functions during data collection, or their buildings were no longer available. Therefore, there were 9831 CHCs analyzed, and the response rate was 99.2%.

Table 1 shows the number of health centers per province. Based on that table, the majority of the CHCs, which included those with inpatient status, were located in rural areas. In Indonesia, at least 2262 health centers were not accredited. Being an unaccredited CHC implied that they had not accomplished accreditation when this research was conducted. Regarding financial management, most were non-public services (non-BLUD), which indicated that the operational funds were entirely from the local government, whereas BLUD refers to CHCs that had satisfied the requirements for some degree of financial autonomy. The highest numbers of health centers were in West and East Java, and the smallest numbers were in Bangka Belitung and North Kalimantan. Table 2 shows the number of health-center applications based on regional criteria.

Table 2 shows disparities in the numbers and percentages of available health applications. In general, urban and rural areas had more health applications, except for E SISMAL, which was more widely available in rural and remotes areas—71.16% and 75.39%, respectively. This pattern was different from urban areas, which did not have many of these applications. Table 3 shows the regional division in Indonesia, which consists of seven regions.

Table 3 shows that many health centers in Indonesia have applications for these programs—more than 50%. Furthermore, ASPAK, PISPK, and P.care were owned by approximately 90% of health centers. As a result, there were some regions, 4, 5, 6, and 7, that had health applications. On average, the health centers had 10 information system programs (mean = 10.37 and median of 10.80). The results have been displayed in the form of a map by region.

In Figure 1, the categories are divided into four quartiles, namely, areas with means of 5.5 to <10.5, 10.5 to <10.7, 10.7 to <10.8, and 10.8 to 10.9. The best areas were region 2, followed by 1 (Sumatra), and then 3 (Nusa Tenggara). Meanwhile, 4, 5, 6, and 7 were in one category, as they had means of 5.5 to <10.5.

In Figure 2, the area is divided based on quartiles, namely, 5.13 to <9.96, 9.96 to <10.78, 10.78 to <11.27, and 11.27 to 11.87. It is shown that in the best region (region 2), not all provinces had a high mean of 11.27–11.87. Those with the highest mean were observed in the Banten and East Java provinces. The highest mean, that of region 1 (Sumatra Island), was discovered in the Jambi, Lampung, and Bangka Belitung provinces. Apart from regions 1 and 2, only Gorontalo Province a mean in the highest quartile. Meanwhile, only South Sulawesi Province had a high mean of 10.78 to <11.27.

Table 4 shows the proportion of CHCs with Information Application or Program by Province. It shows that Papua and West Papua had percentages of less than 60% for all types of information application. Furthermore, Jakarta had less than 50% ownership for several programs, namely, SIHA, SIHEPI, and ESISMAL. Programs with less than 60% ownership in several provinces included SKDR, SIHA, SIHEPI, SIPD3I, and ESISMAL. However, there was no SKDR ownership of more than 90%. Many provinces had below 70% ownership of HFIS, even though this was mandatory for the national health insurance provider. The same was true for SIHA and SIHEPI, many of which still had SIPD3I percentages below 70%.

## 4. Discussion

The analysis showed disparities between the regions and provinces regarding the number of applications and data collection of health programs. Figure 1 indicates that the eastern part of Indonesia is significantly different from the western part. The central part was almost the same as eastern Indonesia regarding the average number of applications owned. Furthermore, some provinces had a small average of applications or programs compared to others. This is common in LMICs where the health information systems are strongly influenced by human-resource factors, institutional funding, foreign aid, corruption, transparency, and poor priorities [5,14].

The province of Jakarta, where the capital city is located, had an average number of applications or health programs which was not high compared to other provinces on the island of Java. Although this has never been analyzed, it is possible, since the CHCs in Jakarta are different from those in other regions. For example, the CHCs in this province are located almost exclusively in sub-districts (kelurahan). However, they are independent in their work and not branch of the CHCs in some large sub-district [15]. Therefore, many sub-districts’ CHCs did not have information-system programs, since they shared tasks with sub-district health centers. Another explanation is that Jakarta was not an endemic area for diseases such as malaria; hence, the use of ESISMAL was deficient, at approximately 25%. In contrast to the disparity problem in eastern Indonesia, there are issues pertaining to the availability of PCs, the electricity supply, workforce availability and their skills, Internet access, etc.

This disparity in one part of the health information system causes various problems. The primary issue is interoperability with security and intellectual property [1,14,16,17,18]. The poor implementation is caused by the absence of monitoring or evaluation [6,19]. Furthermore, its evaluation is complex in many countries, since there are no tools available to assess them, unlike the information system in the maternal sector, which has the MADE IN/OUT method for evaluation [9].

A system for recording health and disease programs can replace extensive surveys and provide good-quality data [4]. They are used in research, even for randomized clinical trials [2,20]. Additionally, some health facilities use information systems capable of diagnosing diseases [21]. Their implementation is related to the competence of human resources and updating programs or applications [22,23,24,25]. However, a lack of human resources and not being consistent with the number of patients will cause the system to not work well and produce poor-quality data [23,25,26].

Support from the government is required to improve the implementation of the health information system [4,27]. The disparities in the health sector among regions in Indonesia exist in the utilization of health facilities, not only in the information system alone [28]. According to the regulation of Minister of Health Number 97 in 2015, the government emphasized the procurement of data on health facilities, data flow, and access. Hence, the problem of interoperability and data quality was not a significant issue [1].

Furthermore, many things should be considered in the health information system, such as electronic medical records, and integration or bridging with other data, especially with population identity data, as attempted in Brazil [17,29], including system modifications for certain groups [30].

A limitation of this research was the absence of an evaluation in the implementation process and the quality of the data output. Furthermore, not all information systems in health centers were assessed, such as medical records, or CHC information systems, such as SIMPUS. In contrast, the inclusion of national data from all CHCs registered in the Health Service and the Ministry of Home Affairs would require a great effort and an enormous cost.

Based on the disparities shown through the results of this study, it is necessary to strengthen the health system in the form of surveillance of data and more significant support from the provincial and district levels to CHCs to ensure the uploading and operation of the required applications. Another suggestion is that the government should be expected to evaluate the information systems of the CHCs, especially regarding data entry for disease-based health programs.

Additionally, interoperability capabilities, deployment of human resources, and updating the system according to the current situation should also be monitored continuously.

Disease data applications involve the digitalization of disease records. The deployment of digital systems requires well-designed applications, supporting and changing management, and strengthening human resources to realize and sustain system health gains [31]. This process does not just happen, but begins with digitization and digital transformation [32]. Nowadays, the digitalization of the system has become an important determinant [33].

## 5. Conclusions

This analysis showed disparities in the health information system in Indonesia by province and region. For example, although the islands of Java, Bali, and Sumatra had attractive situations when broken down, some provinces had poor information systems in their CHCs regarding availability and the number of applications or data-storage programs used. The results of this analysis recommend future improvement of the CHCs’ information systems. The gap between the western and the eastern parts of Indonesia needs to be overcome by increasing resources, including human resources and supportive management.

## Figures and Tables

**Figure 1 ijerph-20-04384-f001:**
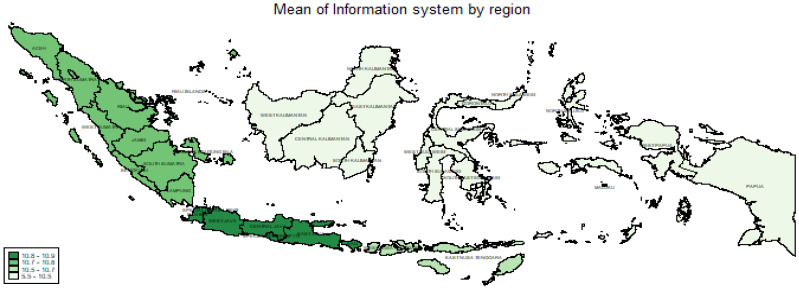
Mean presence of an information system by region.

**Figure 2 ijerph-20-04384-f002:**
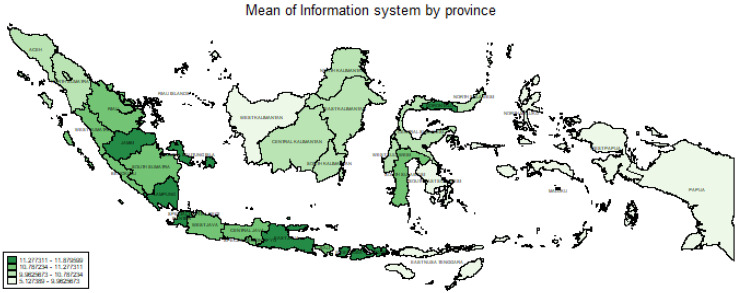
Mean presence of an information systems by province.

**Table 1 ijerph-20-04384-t001:** Characteristics of the CHCs.

	Characteristics	n	%		Province	N CHCs		Province	N CHCs
1	Location of CHC			1	Aceh	347	18	West Kalimantan	241
	- Urban	2447	24.9	2	North Sumatera	571	19	Central Kalimantan	197
	- Rural	4150	42.2	3	West Sumatera	271	20	South Kalimantan	232
	- Remote/Very remote	1946	19.8	4	Riau	216	21	East Kalimantan	178
	- Uncategorized	1288	13.1	5	Jambi	193	22	North Kalimantan	55
2	Service category			6	Bengkulu	151	23	West Nusa Tenggara	161
	- Outpatient	4094	41.6	7	South Sumatera	328	24	East Nusa Tenggara	374
	- Inpatient	5737	58.4	8	Lampung	289	25	North Sulawesi	193
3	Accreditation status			9	Bangka Belitung	55	26	Central Sulawesi	196
	- Basic	2434	24.8	10	Riau Islands	80	27	South Sulawesi	452
	- Madya (Intermediate)	4247	43.2	11	Special Region of Jakarta	313	28	Southeast Sulawesi	281
	- Utama (Advanced)	826	8.4	12	West Java	1069	29	Gorontalo	93
	- Paripurna (Very advanced)	62	0.6	13	Central Java	876	30	West Sulawesi	94
	- No accreditation	2262	23	14	Special Region of Jogjakarta	121	31	Maluku	199
4	Finance management status			15	East Java	964	32	North Maluku	129
	- BLUD	3239	32.9	16	Banten	233	33	West Papua	157
	- Non BLUD	6592	67.1	17	Bali	98	34	Papua	356
	- Total	9831							9831

**Table 2 ijerph-20-04384-t002:** Number of applications or programs in urban, rural, and remote areas.

	Urban		Rural		Remote and Very Remote		*p*
	n	%	n	%	n	%	
Ministry of Health App							
ASPAK							
- Available	2364	96.61	3982	95.95	1622	83.35	0.001
- Not available	83	3.39	168	4.05	324	16.65	
PIS-PK							
- Available	2325	95.09	3925	94.58	1621	20.6	0.001
- Not available	120	4.91	225	5.42	325	48.5	
SKDR							
- Available	1651	67.47	2719	65.52	1041	53.49	0.001
- Not available	796	32.53	1431	34.48	905	46.51	
Health Insurance							
P Care							
- Available	2414	98.65	4058	97.78	1513	77.75	0.001
- Not available	33	1.35	92	2.22	433	22.25	
HFIS							
- Available	1955	79.89	3113	75.01	844	43.37	0.001
- Not available	492	20.11	1037	24.99	1102	56.63	
Health Program App							
SITT							
- Available	2236	91.38	3650	87.95	1384	71.12	0.001
- Not available	211	8.62	500	39.3	562	28.88	
SIHA							
- Available	2243	91.70	3626	87.37	1290	66.29	0.001
- Not available	203	8.30	524	12.63	656	33.71	
SIHEPI							
- Available	1637	66.93	2814	67.81	1039	53.39	0.001
- Not available	809	26.5	1336	43.8	907	29.7	
SIPTM							
- Available	2167	88.56	3611	87.01	1409	72.40	0.001
- Not available	280	11.44	539	12.99	537	27.60	
SIPD3I							
- Available	1952	79.77	3346	80.63	1322	67.93	0.001
- Not available	495	20.23	804	19.37	624	32.07	
ESISMAL							
- Available	1587	64.85	2953	71.16	1467	75.39	0.001
- Not available	860	35.15	1197	28.84	479	18.9	
SISTBM							
- Available	2080	85.00	3594	86.60	1416	72.76	0.001
- Not available	367	15.00	556	13.40	530	27.24	
E PPGBM							
- Available	2175	88.92	3744	90.22	1510	77.60	0.001
- Not available	271	11.08	406	9.78	436	22.40	

**Table 3 ijerph-20-04384-t003:** Numbers and percentages of the presence of information programs based on the number of CHCs in each region in Indonesia.

			Region														
	Indonesia		1		2		3		4		5		6		7		*p*
	n	%	n	%	n	%	n	%	n	%	n	%	n	%	n	%	
Ministry of Health App																	
ASPAK																	
- Available	9103	92.59	2447	96.07	3587	97.05	506	94.58	849	94.02	1237	94.50	232	70.73	245	47.76	0.001
- Not available	728	7.41	100	3.93	109	2.95	29	5.42	54	5.98	72	5.50	96	29.27	268	52.24	
PIS-PK																	
- Available	8945	91.02	2372	93.17	3518	95.24	501	93.64	811	89.81	1173	89.61	266	81.10	304	59.26	0.001
- Not available	883	8.98	174	6.83	176	4.76	34	6.36	92	10.19	136	10.39	62	18.90	209	40.74	
SKDR																	
- Available	6183	62.89	1645	64.59	2530	68.45	338	63.18	545	60.35	784	59.89	154	46.95	187	36.45	0.001
- Not available	3648	37.11	902	35.41	1166	31.55	197	36.82	358	39.65	525	40.11	174	53.05	329	63.55	
Health insurance App																	
P care																	
- Available	9134	92.91	2471	97.02	3678	99.51	521	97.38	845	93.58	1244	95.03	218	66.46	157	30.60	0.001
- Not available	697	7.09	76	2.98	18	0.49	14	2.62	58	6.42	65	4.97	110	33.54	356	69.40	
HFIS																	
- Available	6784	69.01	1824	71.61	3181	86.07	241	45.05	571	63.23	772	58.98	95	28.96	47	9.16	0.001
- Not available	3047	30.99	723	28.39	515	13.93	294	54.95	332	36.77	537	41.02	233	71.04	466	90.84	
Health program App																	
SITT																	
- Available	8333	84.76	2114	83.00	3386	94.75	428	80.00	699	77.41	1081	82.58	233	71.04	276	53.80	0.001
- Not available	1498	15.24	433	17.00	194	5.25	107	20.00	204	22.59	228	17.42	95	28.96	237	46.20	
SIHA																	
- Available	8121	82.6	2114	83.03	3365	91.04	413	77.2	678	75.08	1083	82.73	202	61.59	266	51.85	0.001
- Not available	1709	17.39	432	16.97	331	8.96	122	22.80	225	24.92	226	17.27	126	38.41	247	48.15	
SIHEPI																	
- Available	6223	63.31	1759	69.09	2332	63.10	346	64.67	557	61.68	898	68.60	172	52.44	159	30.99	0.001
- Not available	3607	36.69	787	30.91	1364	36.90	189	35.33	346	38.42	411	31.40	156	47.56	354	69.01	
SIPTM																	
- Available	8223	83.64	2162	84.88	3267	88.39	463	86.54	744	82.39	1116	85.26	239	72.87	281	54.78	0.001
- Not available	1608	16.36	385	15.12	429	11.61	72	13.46	159	17.61	193	14.74	89	27.13	232	45.22	
SIPD3I																	
- Available	7535	76.65	2051	80.53	2938	79.49	400	74.77	665	73.64	1016	77.62	215	65.55	250	48.73	0.001
- Not available	2296	23.35	498	19.47	758	20.51	135	25.23	238	26.36	293	22.38	113	34.45	263	51.27	
ESISMAL																	
- Available	6807	69.24	2042	80.17	1904	51.52	444	82.99	723	80.07	1107	84.57	243	74.09	344	67.05	0.001
- Not available	3024	30.76	505	19.83	1792	48.48	129	17.01	180	19.93	202	15.43	85	25.91	169	32.95	
SISTBM																	
- Available	8090	82.29	2182	85.67	3133	84.77	452	84.49	743	82.28	1135	86.71	219	66.77	226	44.05	0.001
- Not available	1741	17.71	365	14.33	563	15.23	83	15.51	160	17.72	174	13.29	109	33.23	287	55.95	
E PPGBM																	
- Available	8491	86.4	2300	90.30	3285	88.90	466	87.10	775	85.83	1154	88.16	242	73.48	269	52.44	0.001
- Not available	1339	13.62	247	9.70	410	11.10	69	12.90	128	14.17	155	11.84	86	26.22	244	47.56	
Number of Application																	
Mean	10.37		10.79		10.86		10.57		10.19		10.54		8.32		5.77		
Median	11		12		11		11.5		11		12		9.5		6		

**Table 4 ijerph-20-04384-t004:** Proportion of CHCs with Information Application or Program by Province.

		ASPAK		KS		SKDR		PCare		HFIS		SITT		SIHA		SIHEPI		SIPTM		SIPD3I		ESISMAL		SISTBM		EPPGBM		Mean
		n	%	n	%P	n	%	n	%	n	%	n	%	n	%	n	%	n	%	n	%	n	%	n	%	n	%	
		1		2		3		4		5		6		7		8		9		10		11		12		13		
1	Aceh	323	93.1	320	92.2	200	57.6	340	98	236	68.0	252	72.6	257	74.1	245	70.6	271	78.1	257	74.1	247	71.6	280	80.7	294	84.7	10.1
2	North Sumatrera	538	94.2	518	90.7	330	67.4	543	95.1	410	71.8	434	76	429	75.1	315	55.2	443	77.6	409	71.6	372	65.1	470	82.3	499	87.4	10.0
3	West Sumatera	264	97.4	253	93.4	168	62	267	98.5	203	74.9	239	88.2	237	87.5	175	64.6	230	84.9	233	86	237	87.5	237	87.5	261	96.3	11.1
4	Riau	214	99.1	201	93.1	173	80.1	211	97.7	133	61.6	188	87.0	180	83.3	143	66.2	191	88.4	175	81.0	187	96.6	185	85.6	191	88.4	10.9
5	Jambi	191	99.0	183	94.8	139	72.0	189	97.9	152	78.8	167	86.5	160	82.9	158	81.9	174	90.2	169	87.6	177	91.7	182	94.3	184	95.3	11.5
6	South Sumatera	327	99.7	303	92.4	202	61.6	325	99.1	249	75.9	286	87.2	290	88.4	249	75.9	292	89.0	279	85.1	279	85.1	277	84.5	267	90.5	11.1
7	Bengkulu	130	86.1	138	91.4	95	62.9	147	97.4	90	59.6	119	78.8	130	86.1	106	70.2	130	86.1	124	82.1	135	89.4	128	84.8	131	86.8	10.6
8	Lampung	293	98.0	294	98.3	225	75.3	298	99.7	241	80.6	287	96.0	282	94.3	242	87.6	281	94.0	261	87.3	269	90.0	272	91.0	287	96.0	11.9
9	Babel Island	53	96.4	51	92.7	45	81.4	55.0	100	49	89.1	39	70.9	51	92.7	34	61.8	46	83.6	47	85.5	46	83.6	47	85.5	50	90.9	11.1
10	Riau Island	79	98.8	78	97.5	45	56.3	60.0	75	37	46.3	72	90.0	66	82.5	46	57.5	70	87.5	62	77.5	58	72.5	69	86.3	70	87.5	10.1
11	Special Region of Jakarta	275	87.9	193	61.7	198	63.3	312	99.7	251	80.2	274	87.5	136	43.5	129	41.2	255	81.5	183	58.5	79	25.2	228	72.8	248	79.2	8.8
12	West Java	1039	97.2	1043	97.6	763	71.4	1056	98.8	970	81.4	992	92.8	982	91.9	635	59.4	970	90.7	893	83.5	431	40.3	918	85.9	989	92.5	10.8
13	Central Java	859	98.1	863	98.5	594	67.8	873	99.7	772	88.1	850	97.0	854	97.5	563	64.3	777	88.7	711	81.2	553	63.1	771	88.0	797	91.0	11.2
14	Special Region of Jogjakarta	104	99	104	100	75	71.4	105	100	97	92.4	103	98.1	105	100	55	52.4	98	93.3	95	90.5	37	35.2	93	88.6	104	99.0	13
15	East Java	950	98.5	956	99.2	660	68.5	964	100	889	92.2	939	97.4	943	97.8	736	76.3	868	90	781	81	630	65.4	823	85.4	822	85.3	10.4
16	Banten	225	96.6	226	97	145	62.2	232	99.6	200	85.8	215	92.3	211	90.6	132	56.7	179	76.8	167	71.7	89	38.2	177	76.0	196	84.1	10.3
17	Bali	98	100	98	99	70	71.4	98	100	72	73.5	96	98.0	96	98.0	60	61.2	86	87.8	82	83.7	65	66.3	90	91.8	95	96.9	11.3
18	NTB	161	100	160	99.4	110	68.3	161	100	114	70.8	149	92.5	150	93.2	116	72.0	150	93.2	125	77.6	154	95.7	146	90.7	150	93.2	11.5
19	NTT	345	92.2	341	91.2	228	61	360	96.3	180	48.1	279	74.6	263	70.3	230	61.5	313	83.7	275	73.5	290	77.5	306	81.8	316	84.5	9.9
20	West Kalimantan	222	92.1	216	89.6	132	54.8	228	94.6	144	59.8	175	72.6	179	74.3	133	55.2	193	80.1	169	70.1	178	73.9	186	77.2	195	80.9	9.7
21	Central Kalimantan	186	94.4	169	85.8	140	71.1	181	91.9	111	56.3	154	78.2	131	66.5	113	57.4	169	85.8	150	76.1	172	87.3	167	84.4	208	89.7	10.2
22	South Kalimantan	216	93.1	219	94.4	137	59.1	227	97.8	180	77.6	167	72.0	167	72.0	155	66.8	198	85.3	170	73.3	176	75.9	189	81.5	167	84.8	10.4
23	East Kalimantan	168	97.7	153	89	102	59.3	165	95.9	111	64.5	153	89.0	145	84.3	113	65.7	139	80.8	129	75	148	86.0	151	87.8	159	92.4	10.7
24	North Kalimantan	51	92.7	48	87.3	29	52.7	38	69.1	20	36.4	44	80.0	50	90.9	39	70.9	40	72.7	43	78.2	45	81.8	44	80.0	40	72.7	9.6
25	North Sulawes	185	95.9	158	81.9	130	67.4	165	85.5	134	69.4	156	80.8	136	70.5	135	69.9	170	88.1	162	83.9	169	87.6	157	81.3	168	87.0	10.5
26	Central Sulawesi	184	93.9	180	91.8	122	62.2	186	94.9	124	63.3	153	78.1	165	84.2	136	69.4	161	82.1	154	78.6	159	81.1	169	86.2	172	87.8	10.5
27	South Sulawesi	445	98.5	430	95.1	270	59.7	450	99.6	282	62.4	378	83.6	403	89.2	323	71.5	401	88.7	355	78.5	395	87.4	420	92.9	436	96.5	11.0
28	Southeast Sulawesi	242	86.1	229	81.5	134	47.7	262	93.2	135	48.0	230	81.9	209	74.4	170	60.5	206	73.3	199	70.8	217	77.2	223	79.4	199	70.8	9.4
29	Gorontalo	91	97.8	84	90.3	66	71	92	98.9	53	57	84	90.3	85	91.4	70	75.3	91	97.8	78	83.9	86	92.5	85	91.4	88	94.6	11.3
30	West Sulawesi	90	95.7	92	97.9	62	66	89	94.7	44	46.8	80	85.1	85	90.4	64	68.1	87	92.6	68	72.3	81	86.2	81	86.2	91	96.8	11.0
31	Maluku	124	62.3	159	79.9	78	39.2	107	53.8	36	18.1	131	65.8	109	54.8	91	45.7	130	65.3	121	60.8	127	63.8	121	60.8	136	68.3	7.4
32	Nort Maluku	108	83.7	107	82.9	76	58.9	111	86.0	59	45.7	102	79.1	93	72.1	81	62.8	109	84.5	94	72.9	116	89.9	98	76.0	106	82.2	9.8
33	West Papua	89	56.7	70	44.6	31	19.7	62	39.5	11	7.0	75	47.8	77	49.0	32	20.4	62	39.5	66	42.0	106	67.5	59	37.6	65	41.4	5.1
34	Papua	156	43.8	234	65.7	156	43.8	95	26.7	36	10.1	201	56.5	189	53.1	127	35.7	170	47.8	184	51.7	238	66.9	167	46.9	204	57.3	6.0

## Data Availability

The datasets generated during and/or analyzed during the current study are not publicly available due to ethical reasons but are available upon request from the Director of the NIHRD, which has now been renamed the Health Policy Agency of Ministry of Health Indonesia.

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
