# Peer review of "Disparities of Health Program Information Systems in Indonesia: A Cross-Sectional Indonesian Health Facility Research 2019"

_ijerph, 2023, doi:10.3390/ijerph20054384_

Round 1
Reviewer 1 Report
Thank you for giving me the opportunity to read this paper.
I think that there is confusion in the methods section. The authors write about a team of two members.... but then they mention each team... on next line.
Criteria was also mentioned, but criteria for what, for interviewees to be included in the study, or what? Why was the inclusion criteria less than 45 years? What is the role of the enumerator in the study? What is the survey information concisting of?
In section 2.3, the different types of recordings could have been presented in a table.
It is difficult to understand the data collection versus the data management, as data management also includes data collection. What kind of data collection is considered in the two different data collection contexts?
It should be clear what data are used for in different analyses.
The text in the first paragraph on page 7 is recommended to put in a table.
It is not clear what data that come from which data set, already collected electronic data, interview data, survey data etc.
Some of the collected data are missing, as interview data,survey data etc.
The discussion just consist of a compilation of already collected national electronic data. Thus, this paper cannot be considered as a scientific paper.
Several things discussed in chapter 4 are not presented in the results, and should not be discussed in the paper at all.
Author Response
Dear reviewer1. Please find the attachment, the manuscript revised will be sent soon

Reviewer 2 Report
Show the phenomenon and research gap in introduction
Your research looks like case report not article
Im not sure table 2 and table 3 has p value 0.001 can you send the database? or stata output
table 4 its like screenshot from excel and not professional
conclusion too short not answer your research problem
Add references more than 30 and 80% must >2018
Author Response
Dear Reviewer2, please find the attachment. The manuscript revised will be sent soon.

Reviewer 3 Report
The paper addresses an important issue, and provides an interesting analysis of data already collected. However, there are a number of issues with the presentation and analysis of the data that need to be addressed, as well as some further consideration of the discussion of the results.
Abstract
Line 10 ‘aimed to demonstrate the disparity’ implies that disparity exists – investigate would be more appropriate
Introduction
Line 25 ‘evenness’- unclear what is meant here ? equitable distribution
Line 31 ‘imputation’ – do you mean data input ?
Line 34 ‘unavailability of monitoring and evaluation ‘ – perhaps lack of M&E
Line 52 again imputation (which means inference) – I think you mean data entry ? the capacity to enter data ?
Line 56 omit ‘was’ ; replace ‘demonstrate’ with ‘investigate’ – since there is a need to confirm whether there is a disparity in the HIS and its extent
2.2 Data collection
Clarify whether the survey visited all CHC in each district or whether there was a sample of CHCs visited – if a sample, how was the sample determined, what was its size, how were CHC selected to be included in the sample.
2.3 Questionnaire (note misspelling)
It is not clear what information was collected for each of these data applications. For example, whether the application was loaded onto the PC; whether it had ever been opened / used; how much data had been entered; dates of data entry; whether the data had been transferred to district / province / or central level.
Lines 98 ff – again, not clear whether a sample was taken of the different CHCs, or whether all CHCs were visited. According to Pusdatin, there were 10,062 Puskesmas in 2019, indicating that nearly all CHCs were included.
Results
Table 1 and line 132 It would be useful to clarify that BLUD refers to CHC that have satisfied the requirements for some degree of financial autonomy.
Table 2 title and line 136. This table is quite confusing. The title refers to the ‘number of application or program’, but it is unclear how this relates to the number of CHC. It would appear that the total available + not available in each category adds to 8543. This is not consistent with the sample size reported as 9831 (line 125)
In addition, the proportions / percentages of CHC listed in the table seem to be based on the distribution across the urban / rural / remote geographic zones. This would need to be compared to the proportion of the total CHC across the 3 zones. It would be much clearer to compare the proportion of available / not available in each zone, and to provide only one figure (eg the % facilities where the application was not available). Line one would then become not available Urban 3.4%; rural 4.0%; remote 16.6% - clearly indicating that it is the remote that have the highest proportion of CHC without ASPAK. The statistical test used to determine the p level should also be mentioned ? chi squared.
Line 141 refers to E SISMAL as 49.2% available in rural and 26.4% in remote (although the table figure for remote is 24.4%); however, if the proportion is measured as per the total facilities in each region, the figures would be 2953/4150=71.1%; and 1467/1946=75.3%,
Table 3 This table is arranged in the same way as table 2 and has the same difficulty in comparing across regions, and a title that is misleading. The table is based on CHC, so should be the number of CHC with the respective information program in each region. It would be clearer if the percentage “not available” in each column referred to the total CHC in each region. This would indicate, for example that the proportion of CHC without ASPAK in region 7 was 51.5%, much more than the proportion in region 1 (3.9%). This would also be consistent with the calculation for column 1 ‘Indonesia’, and enable comparison between each region and the national average.
Line 146 ff refers to ASPAK and HFIS as being owned by 60% of health centres – although the first column in table 3 titled Indonesia reports that 92.6% of HC have ASPAK, and 69% have HFIS.
Line 152 The division into quartiles is somewhat misleading, as the distribution is not progressive across the quartiles, but 5 regions have means between 10.5 and 10.9; with only 2 regions with a mean below 10.5. Given this skewed distribution, I suggest omitting the map.
Line 156. It is also not appropriate to provide the information that should be in a table, as text, as in line 156 to 164. This is almost incomprehensible for a reader. If this information is relevant to the research question, then it should be included as a table. Given the information presented in tables 2 and 3, I do not think the additional information on mean applications per province contributes to the research question, and the paragraph should be omitted. Similarly, as map 2 closely resembles map 1, it could also be omitted.
Table 4 provides the information of Table 3 in the form that I have suggested for Table 3; that is as the proportion of facilities with the nominated program, but divided by province rather than by region. This information is too detailed for inclusion in the published paper, and could be provided as a supplementary table rather than included in the printed paper.
Discussion
While there is some consideration of potential reasons for the lower number of applications per CHC in Jakarta, there is little consideration of the much more marked disparities in eastern Indonesian provinces. Consideration could be given to issues such as availability of PCs; electricity supply; workforce availability and skills; internet access.
Further consideration could also be given to the significance of the results in terms of health system function; and recommendations to address the disparities, which could include surveillance of the functioning of applications , and greater support from provincial and district level to CHCs to ensure the uploading and operation of the required applications.
Author Response
Dear Reviewer3, please find the attachment. The manuscript revosed will be sent soon.

Round 2
Reviewer 3 Report
Comments on revised version in italics
The authors have largely addressed all the comments and advice provided on the initial version. This has resulted in an improved version that could be publishable.
I have reviewed the clean version of the manuscript and can confirm my decision to accept the manuscript. However, the clean version will still require some editing, mainly in terms of lay out, but also to check spelling (eg questionnaire is misspelt). I assume that this will be addressed during the publication process.
Abstract
Line 10 ‘aimed to demonstrate the disparity’ implies that disparity exists – investigate would be more appropriate
Response: addressed ‘potential’ added
Introduction
Line 25 ‘evenness’- unclear what is meant here ? equitable distribution
Addressed: changed to equitable
Line 31 ‘imputation’ – do you mean data input ?
Addressed: changed to data entry
Line 34 ‘unavailability of monitoring and evaluation ‘ – perhaps lack of M&E
Addressed: changed to lack
Line 52 again imputation (which means inference) – I think you mean data entry ? the capacity to enter data ?
Addressed: changed to data entry
Line 56 omit ‘was’ ; replace ‘demonstrate’ with ‘investigate’ – since there is a need to confirm whether there is a disparity in the HIS and its extent
Partly addressed: still some syntax errors – suggest ‘by investigating information systems, and omit the ‘Demonstrates the disparities is needed to improve health systems’ which is not a complete sentence.
Additional information on the Rifaskes added.
2.2 Data collection
Clarify whether the survey visited all CHC in each district or whether there was a sample of CHCs visited – if a sample, how was the sample determined, what was its size, how were CHC selected to be included in the sample.
Addressed: additional information on data collection method provided
2.3 Questionnaire (note misspelling)
It is not clear what information was collected for each of these data applications. For example, whether the application was loaded onto the PC; whether it had ever been opened / used; how much data had been entered; dates of data entry; whether the data had been transferred to district / province / or central level.
Lines 98 ff – again, not clear whether a sample was taken of the different CHCs, or whether all CHCs were visited. According to Pusdatin, there were 10,062 Puskesmas in 2019, indicating that nearly all CHCs were included.
Addressed – line 72 specifies all CHC in Indonesia were included.
Results
Table 1 and line 132 It would be useful to clarify that BLUD refers to CHC that have satisfied the requirements for some degree of financial autonomy.
Addressed: line 152
Table 2 title and line 136. This table is quite confusing. The title refers to the ‘number of application or program’, but it is unclear how this relates to the number of CHC. It would appear that the total available + not available in each category adds to 8543. This is not consistent with the sample size reported as 9831 (line 125)
In addition, the proportions / percentages of CHC listed in the table seem to be based on the distribution across the urban / rural / remote geographic zones. This would need to be compared to the proportion of the total CHC across the 3 zones. It would be much clearer to compare the proportion of available / not available in each zone, and to provide only one figure (eg the % facilities where the application was not available). Line one would then become not available Urban 3.4%; rural 4.0%; remote 16.6% - clearly indicating that it is the remote that have the highest proportion of CHC without ASPAK. The statistical test used to determine the p level should also be mentioned ? chi squared.
Addressed: Table 2 has been revised to provide % as suggested per zone. The revised table is in accordance with my suggestion as far as I can make out.
Line 141 refers to E SISMAL as 49.2% available in rural and 26.4% in remote (although the table figure for remote is 24.4%); however, if the proportion is measured as per the total facilities in each region, the figures would be 2953/4150=71.1%; and 1467/1946=75.3%,
Addressed; corrected figures provided
Table 3 This table is arranged in the same way as table 2 and has the same difficulty in comparing across regions, and a title that is misleading. The table is based on CHC, so should be the number of CHC with the respective information program in each region.
Addressed: table title changed
It would be clearer if the percentage “not available” in each column referred to the total CHC in each region. This would indicate, for example that the proportion of CHC without ASPAK in region 7 was 51.5%, much more than the proportion in region 1 (3.9%). This would also be consistent with the calculation for column 1 ‘Indonesia’, and enable comparison between each region and the national average.
Addressed: table 3 has been revised to provide the percentages suggested.
Line 146 ff refers to ASPAK and HFIS as being owned by 60% of health centres – although the first column in table 3 titled Indonesia reports that 92.6% of HC have ASPAK, and 69% have HFIS.
Corrected
Line 152 The division into quartiles is somewhat misleading, as the distribution is not progressive across the quartiles, but 5 regions have means between 10.5 and 10.9; with only 2 regions with a mean below 10.5. Given this skewed distribution, I suggest omitting the map.
The authors have decided not to accept this suggestion, and have provided their rationale
Line 156. It is also not appropriate to provide the information that should be in a table, as text, as in line 156 to 164. This is almost incomprehensible for a reader. If this information is relevant to the research question, then it should be included as a table. Given the information presented in tables 2 and 3, I do not think the additional information on mean applications per province contributes to the research question, and the paragraph should be omitted. Similarly, as map 2 closely resembles map 1, it could also be omitted.
Addressed, the paragraph has been omitted.
Table 4 provides the information of Table 3 in the form that I have suggested for Table 3; that is as the proportion of facilities with the nominated program, but divided by province rather than by region. This information is too detailed for inclusion in the published paper, and could be provided as a supplementary table rather than included in the printed paper.
The authors have retained Table 4 as a supplementary table
Discussion
While there is some consideration of potential reasons for the lower number of applications per CHC in Jakarta, there is little consideration of the much more marked disparities in eastern Indonesian provinces. Consideration could be given to issues such as availability of PCs; electricity supply; workforce availability and skills; internet access.
Addressed, additional comments provided.
Further consideration could also be given to the significance of the results in terms of health system function; and recommendations to address the disparities, which could include surveillance of the functioning of applications , and greater support from provincial and district level to CHCs to ensure the uploading and operation of the required applications.
Addressed, additional comments provided.